# Immobilization of an Iridium(I)-NHC-Phosphine Catalyst for Hydrogenation Reactions under Batch and Flow Conditions

**Henrietta Kovács [1,2], Krisztina Orosz [1,2], Gábor Papp [1]** **, Ferenc Joó [1,3]** **and Henrietta Horváth [3],***

[1] Department of Physical Chemistry, University of Debrecen, P.O. Box 400, H-4002 Debrecen, Hungary; kovacshenrietta@science.unideb.hu (H.K.); orosz.krisztina@science.unideb.hu (K.O.); papp.gabor@science.unideb.hu (G.P.); joo.ferenc@science.unideb.hu (F.J.)

[2] Doctoral School of Chemistry, University of Debrecen, P.O. Box 400, H-4002 Debrecen, Hungary

[3] MTA-DE Redox and Homogeneous Catalytic Reaction Mechanisms Research Group, P.O. Box 400, H-4002 Debrecen, Hungary

* Correspondence: henrietta.horvath@science.unideb.hu

**Abstract:** $Na_2[Ir(cod)(emim)(mtppts)]$ (**1**) with high catalytic activity in various organic- and aqueous-phase hydrogenation reactions was immobilized on several types of commercially available ion-exchange supports. The resulting heterogeneous catalyst was investigated in batch reactions and in an H-Cube flow reactor in the hydrogenation of phenylacetylene, diphenylacetylene, 1-hexyne, and benzylideneacetone. Under proper conditions, the catalyst was highly selective in the hydrogenation of alkynes to alkenes, and demonstrated excellent selectivity in C=C over C=O hydrogenation; furthermore, it displayed remarkable stability. Activity of **1** in hydrogenation of levulinic acid to γ-valerolactone was also assessed.

**Keywords:** flow reactor; hydrogenation; immobilization; iridium; N-heterocyclic carbene



## 1. Introduction

In 2013, the production of 25% of pharmaceutical drugs on the market required at least one hydrogenation step [1]. Hydrogenation catalysts with outstanding activity, selectivity, and durability are continuously in demand. Flow chemistry has several benefits for catalytic hydrogenations such as increased safety, improved selectivity, high yields, precise temperature and pressure control, and reduced solvent volumes [2–4].

Homogeneous hydrogenation catalysts are extremely versatile, remarkably selective, and active even under mild conditions [5–7]. However, despite their many advantages, only a limited number of homogeneous catalysts have been successfully commercialized. This is mostly due to the fact that separation of the catalysts from the products poses obvious difficulties in the case of soluble catalysts. For this reason, there has been a renewal of activity in the research of recyclable and robust heterogenized catalysts [8].

Iridium complexes have played the most important role in homogeneous transition metal catalysis [9,10]. They can be used for a large variety of important reactions such as hydrogenation [11–20], hydrosilylation [21], isomerizations [22], isotope exchange [23], carbonylation [24], and biomass conversion reactions, to name a few. Notable examples of industrial applications of iridium-based homogeneous catalysts include the Cativa process for methanol carbonylation [24], and the production of metolachlor by enantioselective imine hydrogenation [25].

Fewer studies have appeared regarding the heterogenization of iridium complexes [26,27] to obtain reusable catalysts despite iridium being an expensive metal and a relatively rare element on Earth. Clearly, the synthesis and catalytic investigation of immobilized iridium complexes is a relevant and important task. In addition, chemical hydrogen storage is also an environmentally important research topic. Several types of iridium-catalyzed hydrogen storage reactions are known in the literature [28–40], however, only a few heterogeneously catalyzed, iridium-based hydrogen storage systems have been described recently [41–43].

One of the possibilities of heterogenization of a soluble metal complex with ionic charge is its reaction with a solid ion-exchanger, as demonstrated by Haag and Whitehurst in their pioneering studies [44]. The support, which may be an inorganic or organic, natural or synthetic material [45–47], may have a large influence on the catalytic properties of the heterogenized metal complex.

Here, we report on the immobilization of the highly active hydrogenation catalyst, Na$_2$[Ir(cod)(emim)(*m*tppts)], **1** (Scheme 1), (cod = 1,5-cyclooctadiene, emim = 1-ethyl-3-methylimidazol-2-ylidene, *m*tppts = *meta*-trisulfonated triphenylphosphine) on commercially available anion-exchange type supports and investigate the stability and catalytic activity of the resulting heterogenized catalysts in simple, organic-phase hydrogenation reactions of alkynes, benzylideneacetone (4-phenyl-but-3-en-2-on), and levulinic acid (4-oxopentanoic acid).

**Scheme 1.** Ir(cod)(emim)(*m*tppts)], **1**, catalyst used in this study.

## 2. Results and Discussion

*2.1. Heterogenization of Na$_2$[Ir(cod)(emim)(mtppts)] (1) and Leaching Tests*

The immobilization of **1** was performed on six different types of commercially available solid ion-exchanger supports. Molselect DEAE-25 and DEAE-Cellulose are crystalline dextrane- and cellulose-based anionites, respectively, functionalized with diethylaminoethyl functional groups. Relisorb QA405/EB has a polymethacrylic matrix, while DIAION HPA25, Amberlite IRA 900, and Lewatit MonoPlus MP 500 are polystyrene-divinylbenzene-based anionites. All four ion exchange resins carry quaternary ammonium functional groups. Macroporous resins have a high effective surface area, and form more rigid beads, making them more resilient under various conditions [47].

All six of the selected anionites were found suitable for anchoring the iridium complex, but there was a significant difference in the individual durations of the immobilization processes, and the stability of the resulting catalysts under acidic and basic conditions. The description of the ion-exchangers and relevant data of immobilization and leaching tests are displayed in Table 1. Figure 1 shows the decrease in time of the absorbance of the Na$_2$[Ir(cod)(emim)(*m*tppts)] solution over Lewatit MonoPlus MP 500.

Immobilization measurements showed that the cellulose- and dextrane-based anionites immobilized the complex very rapidly. However, the resulting heterogenized catalysts performed weakly in the leaching tests, probably due to their tertiary amine functional groups, which are generally less stable under varying pH conditions. Conversely, the synthetic polymer-based supports, especially those with a crosslinked polystyrene matrix, immobilized the complex less readily, but still successfully. Satisfyingly, the macroporous Lewatit MonoPlus MP 500 ion-exchange resin-supported catalyst showed remarkable stability under acidic and basic conditions, with only 1% leaching of the complex in 0.1 M H$_3$PO$_4$ solution and 4% leaching in 0.1 M Na$_3$PO$_4$. The macroreticular Amberlite IRA 900 also performed well in the tests, only slightly falling behind Lewatit MonoPlus MP 500.

**Table 1.** Heterogenization of Na$_2$[Ir(cod)(emim)($m$tppts)] (1) on different types of anion exchangers and results of acid-base leaching test.

| Type of Support | Name | Time [a] (min) | Leaching [b] (%) | |
|---|---|---|---|---|
| | | | 0.1 M H$_3$PO$_4$ | 0.1 M Na$_3$PO$_4$ |
| Cellulose | DEAE-Cellulose | 4 | 36 | 45 |
| Dextrane | Molselect DEAE-25 | 4 | 15 | 27 |
| Polymethacrylate | Relisorb QA405/EB | 8 | 16 | 23 |
| PS/DVB | DIAION HPA25 | 12 | 4 | 8 |
| PS/DVB | Amberlite IRA-900 | 20 | 5 | 4 |
| PS/DVB | Lewatit MonoPlus MP 500 | 32 | 1 | 4 |

[a] Time for complete anchoring, [b] Room temperature, 1 h.

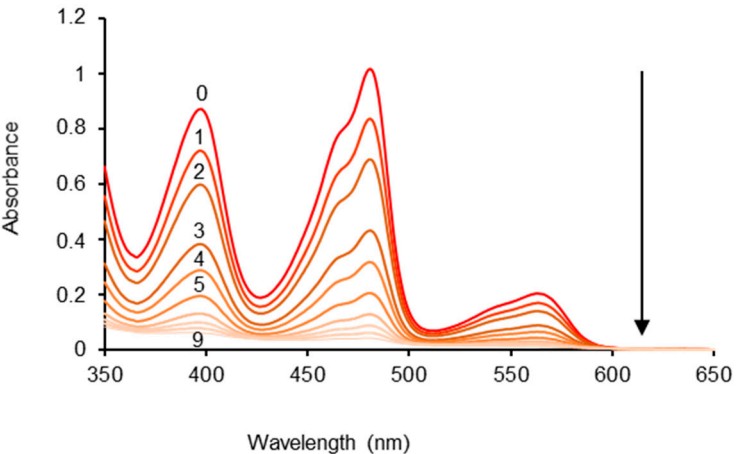

**Figure 1.** Spectrophotometric study of the immobilization of Na$_2$[Ir(cod)(emim)($m$tppts)] (**1**) on Lewatit MonoPlus MP 500 in 0.01 M HCl solution. $T$ = r.t., m(complex) = 1.5 mg, m(resin) = 50 mg, V(0.01 M HCl) = 1 mL, $t_{total}$ = 32 min. Spectra: 0—complex; 1—complex + resin, immediately after mixing; 2–9—taken every 4 min.

Na$_2$[Ir(cod)(emim)($m$tppts)]@Lewatit MonoPlus MP 500 (**1**@Lewatit) has the physical appearance of red beads due to the deposition of red Na$_2$[Ir(cod)(emim)($m$tppts)] onto the translucent yellow ion-exchange resin (Figure S1). Figure S2 shows the infrared spectra of **1**@Lewatit (a) and that of the resin itself (b). The strong absorption of the sulfonate group at 1034 cm$^{-1}$ clearly shows the presence of the Ir-complex bound to the resin.

### 2.2. Reaction of 1@Lewatit with Hydrogen

Keeping the solid Na$_2$[Ir(cod)(emim)($m$tppts)] complex in an H$_2$ atmosphere of 1 bar pressure at room temperature caused a slow color change from red to yellow, and the increase of the absorbance at 2249 cm$^{-1}$ characteristic for Ir–H terminal hydride complexes (see Figure S3). All these spectral changes agree well with the previously published solution behavior of [Ir(cod)(NHC)($m$tppts)] complexes [22]. The phenomenon of yellow coloration also occurs with the supported complex, the red beads turn yellow under 1 bar H$_2$ atmosphere. The formation of the yellow hydride complex can be a consequence of the hydrogenation, and thus the dissociation of the 1,5-cyclooctane ligand from the coordination sphere. This hypothesis is supported by GC-MS analysis of the reaction mixtures after homogeneous and heterogeneous phase hydrogenation, in which the hydrogenated product cyclooctane appears at m/z = 112.4 (Figure S4).

### 2.3. Catalytic Hydrogenation Reactions with 1@Lewatit Supported Catalyst

The activity and selectivity of **1**@Lewatit were evaluated in the hydrogenation of alkynes and an α,β-unsaturated ketone in flow reactions. These experiments were preceded by a comparison of the properties of the soluble and the heterogenized Ir(I)-catalysts in

batch reactions. In addition, hydrogenation of levulinic acid to yield γ-valerolactone was also studied in batch reactions.

### 2.3.1. Hydrogenations with Soluble and Heterogenized Na₂[Ir(cod)(emim)(*m*tppts)] in Batch Reactions

Na₂[Ir(cod)(emim)(*m*tppts)] (**1**) is an active catalyst of the hydrogenation of phenylacetylene and 1-hexyne in a homogeneous methanolic solution (Table 2) under mild conditions. Phenylacetylene underwent 100% conversion already at 50 °C, while at the same temperature, 1-hexyne was hydrogenated only to 90% extent. At or below 20 °C, the reaction of both substrates yielded mostly (>82%) the semi-hydrogenated products styrene and 1-hexene, respectively. The highest selectivity was observed at 20 °C (Table 2, Entries 2 and 8), however, at higher temperatures (and higher conversions), the selectivity was lost and the fully hydrogenated products (ethylbenzene, hexane) were produced. Time course of phenylacetylene hydrogenation at 20 °C showed (Figure S5) that in the first 60 min, the concentration of styrene in the product mixture changed only slowly from 84% to 75% (with simultaneous increase of ethylbenzene concentration), however, following complete hydrogenation of the alkyne, fast hydrogenation of styrene commenced (40% ethylbenzene at 90 min). This finding revealed that in the presence of the alkyne substrate, further hydrogenation of its first reduced product (styrene) was retarded, although complete inhibition of styrene hydrogenation did not occur. Such phenomena are often observed in the hydrogenation of alkynes and are explained by stronger binding of the alkyne to the catalyst relative to coordination of the alkene product [6,48]. From the data of Figure S6, a TOF = 100 h⁻¹ (TOF = turnover frequency = mol reacted substrate x (mol catalyst × time)⁻¹) can be calculated, showing that the catalyst is highly active even at 20 °C.

**Table 2.** Hydrogenation of phenylacetylene and 1-hexyne catalyzed homogeneously by Na₂[Ir(cod)(emim)(*m*tppts)] (**1**), and heterogeneously by the supported catalyst, **1**@Lewatit, in batch reactions.

| Entry | Catalyst [a] | Substrate | T/°C | Conversion | Products [b] | |
|---|---|---|---|---|---|---|
| | | | | % | A/% | B/% |
| 1 | H | Phenylacetylene | 5 | 74 | 75 | 25 |
| 2 | H | Phenylacetylene | 20 | 82 | 85 | 15 |
| 3 | H | Phenylacetylene | 50 | 100 | 0 | 100 |
| 4 | H | Phenylacetylene | 80 | 100 | 0 | 100 |
| 5 | S | Phenylacetylene | 20 | 35 | 75 | 25 |
| 6 | S | Phenylacetylene | 50 | 90 | 85 | 15 |
| 7 | S | Phenylacetylene | 80 | 100 | 10 | 90 |
| 8 | H | 1-Hexyne | 20 | 42 | 82 | 18 |
| 9 | H | 1-Hexyne | 50 | 90 | 23 | 77 |
| 10 | H | 1-Hexyne | 80 | 93 | 4 | 96 |
| 11 | S | 1-Hexyne | 50 | 10 | 82 | 18 |
| 12 | S | 1-Hexyne | 80 | 82 | 55 | 45 |

Conditions: n(substrate) = 1.0 mmol; n(catalyst) = 0.01 mmol; P(H₂) = 10 bar; solvent: methanol; V = 1.0 mL; reaction time 1.0 h. [a] Catalyst: Na₂[Ir(cod)(emim)(*m*tppts)] (**1**) in homogeneous solution (H), or **1**@Lewatit in suspension (S). [b] Relative concentrations of products in the final reaction mixture, %. A: styrene (1–7), 1-hexene (8–12); B: ethylbenzene (1–7), hexane (8–12).

The supported catalyst, **1**@Lewatit, showed very similar catalytic behavior (Figure S6). The reaction rates were somewhat lower than in the homogeneous solutions (e.g., Table 2, Entries 5 vs. 2, 11 vs. 9), most probably due to diffusion limitations. Under optimum conditions, >82% selectivities toward semi-hydrogenation could be achieved (Entries 6 and 11), however, at higher temperatures, reduction to the saturated products took priority.

It is important to note that upon admission of H₂ into the reactors, the color of the catalyst solution or the catalyst beads immediately turned yellow, and this prevailed

during the reactions. The color of the homogeneous reaction mixture did not change after achieving 100% conversion and the same was also observed with the suspended catalyst.

Recycling of the supported catalyst in batch reactions was not investigated, however, in flow reactions, it proved stable for long reaction times (Section 2.3.2).

Previously, we have observed that water had a very peculiar effect on the rate of transfer hydrogenation reactions catalyzed by $Na_2[Ir(cod)(emim)(mtppts)]$ [49]. Namely, the rate of hydrogen transfer reduction of acetophenone in aqueous 2-propanol mixtures showed a significant increase with increasing water concentrations (sometimes upon passing through a minimum at around $x = 0.7$ concentration of 2-propanol; $x$ = mole fraction). Interestingly, with the same catalyst, the same phenomena were also observed in the hydrogenation of phenylacetylene and diphenylacetylene, both with the homogeneous and heterogeneous catalysts (Figures S7–S10).

Earlier, we found that a closely similar water-soluble catalyst, $Na_2[Ir(cod)(bmim)(mtppts)]$ (bmim = 1-butyl-3-methylimidazol-2-ylidene) actively catalyzed the hydrogenation of levulinic acid (LA) to γ-valerolactone (GVL) (Scheme 2) in aqueous solution under mild conditions (atmospheric pressure, 60 °C) [22]. Since this reaction makes possible the production of the highly promising green solvent, GVL, from biomass-based sources [50], we attempted its catalysis by using both the soluble $Na_2[Ir(cod)(emim)(mtppts)]$ (**1**) and **1**@Lewatit (Figure S11). Indeed, in an aqueous solution at 70 °C under 10 bar $H_2$ pressure, 2 mol% of **1** led to the formation of GVL with 26% (1 h), and 47% (2 h) yield. Under the same conditions but with **1**@Lewatit, the yield of GVL was 32% in 2 h and 44% in 4 h. These yields corresponded to TOFs 13.0 (1 h) and 11.8 $h^{-1}$ (2 h) for the homogeneous systems and 8.0 $h^{-1}$ (2 h) and 5.5 $h^{-1}$ (4 h) for the heterogeneous catalyst. Although these activities were modest, these are the first examples of the use of Ir-NHC-phosphine catalysts for the hydrogenation of levulinic acid to GVL.

**Scheme 2.** Hydrogenation of phenylacetylene (**A**), diphenylacetylene (**B**), benzylideneacetone (**C**), and levulinic acid (**D**). (LA: levulinic acid, GVL: γ-valerolactone).

Altogether, the similar spectral changes (Section 3.2) and the characteristic common catalytic properties of $Na_2[Ir(cod)(emim)(mtppts)]$, **1**, and **1**@Lewatit strongly point to the same molecular species acting as a catalyst in both the homogeneous solution and on the surface of the support.

### 2.3.2. Hydrogenations under Flow Conditions

Phenylacetylene, diphenylacetylene, 1-hexyne, and benzylideneacetone (4-phenyl-but-3-en-2-on) were hydrogenated with the supported $Na_2[Ir(cod)(emim)(mtppts)]$ catalyst in an H-Cube flow reactor. The general observations are summarized in Table 3. All experiments were done in the single-pass mode (i.e., the conversions were calculated from the substrate concentration change on passing through the reactor only once).

**Table 3.** Hydrogenation of various alkynes and benzylideneacetone catalyzed by **1**@Lewatit in the H-Cube flow reactor.

| Entry | Substrate | $T/°C$ | Conversion | Products [a] | |
|---|---|---|---|---|---|
| | | | | **A/%** | **B/%** |
| 1 | Phenylacetylene | 60 | 22 | 78 | 22 |
| 2 | Phenylacetylene | 80 | 47 | 80 | 20 |
| 3 | Diphenylacetylene [b] | 60 | 16 | 80 | 20 |
| 4 | Diphenylacetylene [b] | 80 | 34 | 82 | 18 |
| 5 | 1-Hexyne | 60 | 13 | 55 | 45 |
| 6 | 1-Hexyne | 80 | 38 | 57 | 43 |
| 7 | Benzylideneacetone [c] | 60 | 37 | 92 | 7 |
| 8 | Benzylideneacetone [c] | 80 | 48 | 93 | 6 |

Conditions: c(substrate) = 0.05 M; m(catalyst) = 200 mg; $P(H_2)$ = 20 bar; v = 1.0 mL/min; solvent: methanol (1,2,5,6), or toluene (3,4,7,8). [a] Relative concentrations of products in the effluent reaction mixture, %. A: styrene (1,2), *cis*-stilbene (3,4), 1-hexene (5,6), and 4-phenylbutan-2-one (7,8), B: ethylbenzene (1,2), *trans*-stilbene (3,4), hexane (5,6), and 4-phenyl-butan-2-ol (7,8); [b] No diphenylethane; [c] 4-Phenyl-but-3-en-2-ol = 1%.

The data in Table 3 show that the supported Ir(I)-catalyst also retained its activity under flow conditions. With phenylacetylene, an approximately 8:2 selectivity in favor of the semi-hydrogenated product (styrene) was observed, which slightly increased with increased conversions. In contrast, in the hydrogenation of 1-hexyne, substantial overreduction to hexane occurred. No diphenylethane was detected in the product mixture of diphenylacetylene hydrogenation. In addition to complete chemoselectivity, the hydrogenation of diphenylacetylene also showed pronounced selectivity to the formation of *cis*-stilbene (Table 3, Entries 3,4). Control experiments under hydrogenation conditions ruled out cis-to-trans (or vice versa) isomerization of stilbenes.

The hydrogenation of benzylideneacetone was highly selective to the C=C double bond, with overwhelming formation of the saturated ketone (4-phenylbutan-2-one) (Table 3, Entries 7,8); the enol product was detected in ≤1% ratio. The selectivity toward the formation of saturated ketone was also observed with $Na_2[Ir(cod)(emim)(mtppts)]$ in homogeneous solutions. At a temperature of 80 °C, under 10 bar $H_2$, [S]/[C]=100, **1** catalyzed the hydrogenation of benzylideneacetone with 58% total conversion and with 54% yield (93% selectivity) of phenylbutan-2-one together with 4% yield of 4-phenylbutan-2-ol (7%). (Note, that due to the insolubility of **1** in toluene, this reaction has to be carried out in methanol as the solvent.)

The reaction rates showed a pronounced temperature dependence (Figure S12) but did not significantly respond to changes of $P(H_2)$ (Figure S13).

Figure 2 shows the conversion of phenylacetylene as well as the product distribution as a function of catalyst time on stream during hydrogenation. At the beginning of the measurement, the single-pass conversion of phenylacetylene was 46%, which corresponds to a turnover frequency of the catalyst TOF = 25.7 $h^{-1}$.

In relation to the TOF = 100 $h^{-1}$ determined for the analogous homogeneous reaction (Section 3.3), this decreased activity shows the effect of the heterogeneous conditions (diffusion limitations). As can be seen from Figure 2, the original 46% conversion decreased to 24% in 14 h of continuous operation of the reactor, while the styrene:ethylbenzene ratio remained a constant 80:20 after about 7 h (74:26 at the outset). A similar drop in conversion was observed by Horvath et al. using an ion-exchanger supported $[\{RuCl_2(mtppms)_2\}_2]$ catalyst (mtppms = meta-monosulfonated triphenylphosphine) [51] with a flow reactor operated in impulse mode. In our case, in the 840 min time of operation, 420 mL of 0.05 M

phenylacetylene solution flowed through and was hydrogenated by the supported iridium catalyst. During this time, a 22% drop of single-pass conversion was observed, but the catalyst still retained notable catalytic activity.

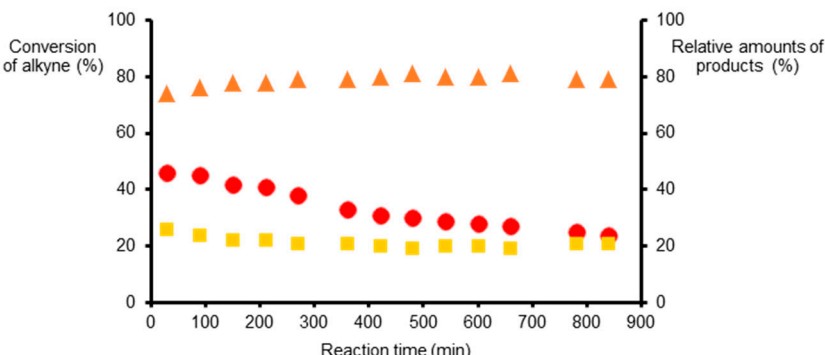

**Figure 2.** Conversion of phenylacetylene (●) and the ratio of styrene (▲) and ethylbenzene (■) in hydrogenation catalyzed by Na$_2$[Ir(cod)(emim)(*m*tppts)]@Lewatit MonoPlus MP 500 in an H-Cube flow reactor as a function of the catalyst time on stream. c(alkyne) = 0.05 M, m(catalyst) = 200 mg, $P$(H$_2$) = 20 bar, $T$ = 60 °C, v = 0.5 mL/min, solvent: methanol.

With the purpose of investigating the stability and recyclability of the supported catalyst, the same reaction was repeated after two weeks of regular use of the same catalyst in the reactor, and finally, six months after the very first use. Impressively, the immobilized catalyst retained a substantial portion of its catalytic activity, and even after six months, it could perform further hydrogenation reactions, which is remarkable for a heterogenized complex catalyst (Figure 3).

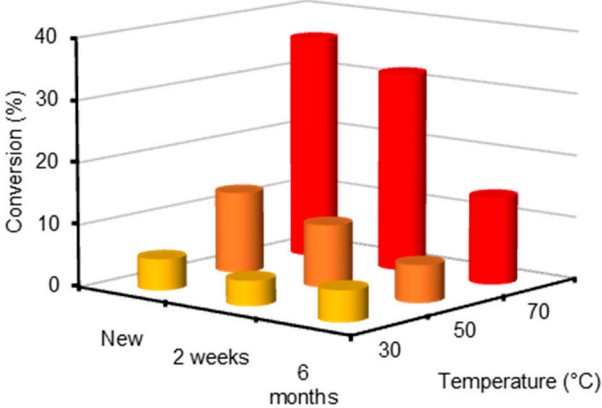

**Figure 3.** Catalytic hydrogenation of phenylacetylene by **1**@Lewatit in an H-Cube flow reactor. c(alkyne) = 0.05 M, m(catalyst) = 200 mg, $P$(H$_2$) = 20 bar, v = 1.0 mL/min, solvent: methanol.

The pronounced stability of the **1**@Lewatit catalyst under hydrogenation conditions is a most welcome feature in comparison to the cases of its soluble analogs. Buriak et al. made an extensive study of the stability of various [Ir(cod)(NHC)(phosphine)X] type catalysts, and showed that these complexes lost their activity at the end of the hydrogenation of the first batch of alkenes [52,53]. After thorough scrutiny of inactivation, they concluded that it was caused by the formation of hydride-bridged iridium dimer and trimer complexes and clusters. Similar inactivation phenomena were also observed in the hydrogenation of water-soluble alkenoic acids in aqueous solutions, which could be largely prevented by applying strongly coordinating bidentate ligands (e.g., oxalic acid)[22]. We assume that site isolation of the Na$_2$[Ir(cod)(emim)(*m*tppts)] molecules on the surface of the ion-exchanger support prevents the formation of Ir(I)-hydride clusters and leads to long maintained

catalytic activity. Such a positive effect of surface site isolation has already been reported in the case of phthalocyanine-ligated metal complex oxidation catalysts [54].

## 3. Materials and Methods

All reagents and solvents were the highest commercially available purity, used without further purification. Alkynes, benzylideneacetone, and the support materials were products of Sigma-Aldrich, City, State Abbr (if has)., Country. $Na_2[Ir(cod)(emim)(mtppts)]$ was synthesized according to the procedure in the literature [22]. Its purity was checked by $^1H$, $^{13}C$, and $^{31}P$ NMR spectroscopy. Manipulations with air-sensitive compounds were carried out using Schlenk techniques.

Details of equipment used in this study are found in the Supplementary Materials.

### 3.1. Anchoring of Na2[Ir(cod)(emim)(mtppts)] on Solid Supports

Anion-exchange support materials were pretreated before heterogenization: these were stirred three times alternatingly in aqueous 0.1 M NaOH and 0.1 M HCl, finally washed with distilled water three times, and dried in vacuo.

In a Schlenk tube, $Na_2[Ir(cod)(emim)(mtppts)]$ (60 mg) was dissolved in deoxygenated 0.01 M HCl solution (12 mL) under an argon atmosphere and the solution was added onto 360 mg of pretreated Lewatit MonoPlus MP 500 ion-exchange beads under an argon atmosphere. The reaction mixture was thermostated at 40 °C for 3 h, during which the solution decolorized, and the beads turned red. The heterogenized complex was filtered under argon, washed with hexane, and dried in vacuo. Yield: 390 mg supported catalyst (96%).

### 3.2. Spectrophotometric Study of the Immobilization Process and Leaching Tests

Heterogenization of $Na_2[Ir(cod)(emim)(mtppts)]$ was investigated at room temperature in an argon-filled inert spectrophotometric cell. A 1.5 mg solid complex and 50 mg solid support were placed into the cuvette, placed under argon, then 1 mL of deoxygenated 0.01 M HCl solution was injected through the septum of the cell. An absorption spectrum was obtained every 4 min while shaking the cuvette between measurements until the absorbance decreased to zero.

For leaching tests, 30 mg of the heterogenized catalyst was placed into a septum-closed vial, flushed with argon, and then 2 mL of 0.1 M $H_3PO_4$ or $Na_3PO_4$ was injected through the septum. Each vial was shaken for an hour at room temperature, and then the UV–VIS absorption spectrum was recorded. The mass of the leached complex in the solution was calculated using a calibration curve (Figure S14).

### 3.3. Catalytic Reactions

The reactions were performed in an H-Cube hydrogenation microreactor (ThalesNano Nanotechnology Inc., Budapest, Hungary). A total of 200 mg heterogenized catalyst was loaded into 30 × 4 mm CatCart stainless steel tubes. In the continuous mode of operation, the substrate solution (0.05 M phenylacetylene or 1-hexyne in methanol, 0.05 M diphenylacetylene or benzylideneacetone in toluene) flowed through the catalyst bed without interruption. Samples were taken regularly at the outflow. Reaction temperatures ranged between 20–80 °C, pressure between 20–80 bar, and flow rate was 0.5–1.0 mL/min. Reaction mixtures were analyzed by gas chromatography; for details, see the Supplementary Materials.

## 4. Conclusions

$Na_2[Ir(cod)(emim)(mtppts)]$, an excellent iridium-NHC-phosphine type hydrogenation catalyst was found to be suitable for anchoring on various types of anion-exchange supports; the process is efficient and inexpensive. $Na_2[Ir(cod)(emim)(mtppts)]$ immobilized on the commercially available Lewatit MonoPlus MP 500 anion-exchange resin proved to be an active, selective, and robust catalyst—both in batch and flow conditions—in the

semi-hydrogenation of phenylacetylene as well as of diphenylacetylene. Notably, hydrogenation of diphenylacetylene yielded *cis*-stilbene with 82% selectivity; diphenylethane, the fully hydrogenated product was completely absent from the reaction mixtures. The supported Ir-NHC-phosphine complex also showed potential to be a selective C=C bond hydrogenation catalyst in the presence of carbonyl groups, which was demonstrated in the hydrogenation of benzylideneacetone. The hydrogenation of the C=C bond was viable without affecting the C=O group, and supplied 4-phenyl-butan-2-one with 93% selectivity. Both the soluble and heterogenized **1** was able to catalyze the hydrogenation of levulinic acid to γ-valerolactone. Comparison of the physical and chemical properties of Na$_2$[Ir(cod)(emim)(*m*tppts)] in the dissolved and heterogenized form suggests that the same molecular species is operating in the homogeneous solution and on the surface of the support.

**Supplementary Materials:** The following are available online at https://www.mdpi.com/article/10.3390/catal11060656/s1, Figure S1: Calibration curve for spectrophotometric leaching tests; Figure S2: Optical microscopy photographs of Lewatit MonoPlus MP 500 and **1**@ Lewatit MonoPlus MP 500; Figures S3–S4: Infrared spectra of Lewatit MonoPlus MP 500 and **1**@ Lewatit MonoPlus MP 500; Figure S5: GC-MS detection of cyclooctane formed in the hydrogenation of **1**; Figures S6–S7: Time course of hydrogenation of phenylacetylene by **1** and **1**@ Lewatit MonoPlus MP 500 in batch reactions; Figures S8–S11: Effect of water on hydrogenation of phenylacetylene and diphenylacetylene by **1** and **1**@ Lewatit MonoPlus MP 500 in batch reactions; Figure S12: Temperature dependence of conversion of diphenylacetylene catalyzed by **1**@ Lewatit MonoPlus MP 500 in the flow reactor; Figure S13: Pressure dependence of the conversion of diphenylacetylene catalyzed by **1**@ Lewatit MonoPlus MP 500 in the flow reactor; Equipments and methods used in the study; Figure S14: Calibration curve for spectrophotometric leaching tests.

**Author Contributions:** Conceptualization, H.H. and F.J., Methodology, G.P.; Synthesis and characterization of catalysts, H.K., K.O., and H.H.; Catalysis experiments, H.K., K.O., and H.H.; Discussion of experimental results, All authors; Writing—Original Draft Preparation, All authors; Writing—Review and Editing, H.K., H.H., and F.J. All authors have read and agreed to the published version of the manuscript.

**Funding:** The research was supported by the EU and co-financed by the European Regional Development Fund (under the projects GINOP-2.3.2-15-2016-00008, GINOP-2.3.2-15-2016-00041, and GINOP-2.3.3-15-2016-00004), and by the Thematic Excellence Program (TKP2020-NKA-04) of the Ministry for Innovation and Technology in Hungary. The financial support of the Hungarian National Research, Development, and Innovation Office (FK-128333) is greatly acknowledged.

**Data Availability Statement:** The data presented in this study are openly available in [repository name e.g., FigShare] at [doi], reference number [reference number].

**Acknowledgments:** The authors are grateful to István Csarnovics (U. Debrecen, Dept. Experimental Physics) and István Szabó (U. Debrecen, Dept. Solid State Physics) for their help in catalyst characterization and for useful discussions.

**Conflicts of Interest:** The authors declare no conflict of interest.

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
