# Peer review of "Immobilization of an Iridium(I)-NHC-Phosphine Catalyst for Hydrogenation Reactions under Batch and Flow Conditions"

_catalysts, doi:10.3390/catal11060656_

Round 1
Reviewer 1 Report
The present paper entitled
"Immobilization of an Iridium(I)-NHC-phosphine catalyst for hydrogenation reactions under batch and flow conditions"
by Henrietta Horvath deals with the immobilisation of the Ir-based anionic catalyst Na2[Ir(cod)(emim)(mtppts)] (1) on ammonium-based anion exchangers. Lewatit MonoPlus MP 500 ion-exchange resin gave excellent results (including leaching) and 1@Lewatit was submitted to catalytic test reactions. The unsaturated substrates for hydrogenation test reactions were chosen reasonably and with relevance. The work has been carried out with a high scientific standard. I am happy to recommend this study for publication and only have minor comments:
i) Some formatting should be done: tables should be centred, caption and figure on the same page....
ii) Figure 1. The spectra should be labelled with time they were taken at, e.g. 5 min, 10 min...
iii) The selectivity of C=C vs. C=O hydrogenation in benzylideneacetone for 1@Lewatit is high. Does the non-immobilised catalyst 1 in homogenous solution display the same selectivity?
Reviewer 2 Report
Horváth et. al in their article presented immobilization of an Iridium(I)-NHC-phosphine catalyst for hydrogenation reactions under batch and flow conditions. This is a continuation of previous studies concerning catalyst immobilization and its utilization in hydrogenation or related transformation. In my opinon the presented work is suitable for the publication in the Catalysts after major revision. However, some issues shloud be calrified.
1) Author's used the heterogenous catalysts in batch and flow operations. Did any attempts for the recycling of the catalysts under batch mode were applied?
2) Why did the Authors use spectrophotometric study for the leaching test? The control of the Ir in the final products by ICP-MS or ICP-OES analyzes will better show the leaching during the reaction. If it is possible please add this analysis.
Round 2
Reviewer 2 Report
Accept in present form